# Recognition of Arboviruses by the Mosquito Immune System

**DOI:** 10.3390/biom13071159

**Published:** 2023-07-21

**Authors:** Brian C. Prince, Elizabeth Walsh, Tran Zen B. Torres, Claudia Rückert

**Affiliations:** Department of Biochemistry and Molecular Biology, College of Agriculture, Biotechnology & Natural Resources, University of Nevada, Reno, NV 89557, USA; brianprince@unr.edu (B.C.P.); elizabethwalsh@unr.edu (E.W.); tranzent@nevada.unr.edu (T.Z.B.T.)

**Keywords:** arboviruses, virus recognition, mosquito immunity, PRR, PAMP

## Abstract

Arthropod-borne viruses (arboviruses) pose a significant threat to both human and animal health worldwide. These viruses are transmitted through the bites of mosquitoes, ticks, sandflies, or biting midges to humans or animals. In humans, arbovirus infection often results in mild flu-like symptoms, but severe disease and death also occur. There are few vaccines available, so control efforts focus on the mosquito population and virus transmission control. One area of research that may enable the development of new strategies to control arbovirus transmission is the field of vector immunology. Arthropod vectors, such as mosquitoes, have coevolved with arboviruses, resulting in a balance of virus replication and vector immune responses. If this balance were disrupted, virus transmission would likely be reduced, either through reduced replication, or even through enhanced replication, resulting in mosquito mortality. The first step in mounting any immune response is to recognize the presence of an invading pathogen. Recent research advances have been made to tease apart the mechanisms of arbovirus detection by mosquitoes. Here, we summarize what is known about arbovirus recognition by the mosquito immune system, try to generate a comprehensive picture, and highlight where there are still gaps in our current understanding.

## 1. Introduction

Mosquitoes transmit the causative agents for many human infectious diseases that result in significant morbidity and mortality globally, especially throughout tropical and subtropical regions. These insect vectors transmit many medically important arthropod-borne viruses (arboviruses) belonging to a wide variety of virus families, with the majority of arboviruses belonging to *Bunyavirales*, *Flaviviridae*, and *Togaviridae* [1]. Depending on the virus, symptomatic arbovirus infections may result in high fever, joint pain, encephalitis, hemorrhagic fever, coma, seizures, and sometimes death [2]. Arboviral diseases have undergone significant emergence and resurgence over the last 20 years, including epidemics of chikungunya virus (CHIKV) and Zika virus (ZIKV) [3]. Although this is the result of a variety of factors, anthropogenic impacts are considered to be primary drivers for the increasing incidence of arbovirus infections. These catalysts include rapid growth in global population, excessive urbanization, the disruption of natural habitats, booming international travel, and climate change [4,5]. Controlling mosquito populations as well as virus transmission itself are major strategies to reduce the burden of arboviral diseases. In order to develop novel strategies to interrupt arbovirus transmission, there is a strong need to further elucidate the interactions between arboviruses and their mosquito vectors.

Arboviruses must be highly efficient at crossing the barriers present in their mosquito vectors to be transmitted [6]. Following a viremic blood meal, viruses have to pass through the midgut infection barrier to propagate in the midgut epithelium [6]. The viruses must then escape from the midgut tissue and systemically replicate in other tissues. Finally, viruses need to spread to and replicate in the salivary glands. The following blood feed can then transfer infectious saliva to a new host [6]. The natural transmission cycles of arboviruses rarely result in any discernible loss in fitness or behavioral changes to their mosquito vectors. Viruses tend to persist throughout the lives of infected mosquitoes, which allows for efficient viral transmission [7]. Numerous studies have suggested that the activation of cellular and humoral immune responses limit the replication of arboviruses in mosquitoes in vitro and in vivo (reviewed in [7,8,9,10]). Yet, one of the least understood aspects of antiviral immunity in mosquitoes is how viruses are initially detected by the immune system. 

Innate immune responses are activated by pattern recognition receptors (PRRs), which recognize pathogen-associated molecular patterns (PAMPs) [11,12]. Mosquito antiviral immune responses (Figure 1) include several conserved signal transduction pathways that lead to antiviral effector gene expression and restrict arboviral replication. These pathways include the Janus kinase–signal transducer and activator of transcription (JAK/STAT), Toll, Mitogen-activated protein kinases (MAPK), and immune deficiency (Imd) pathways. While these transcriptional responses have been shown to limit viral replication in some contexts, the RNA interference (RNAi) pathway is considered the central antiviral mechanism in mosquitoes [13,14]. In addition to these pathways, mosquitoes possess other defenses, such as phagocytosis and melanization [15]. In most situations, PRRs are responsible for either starting signaling cascades or acting as opsonins. Any detailed understanding of PRRs and virus sensing is still quite scarce in mosquitoes, and knowledge obtained from other dipterans, such as *Drosophila melanogaster*, is often used as the most likely indicator for mosquito immune mechanisms. However, mosquitoes vary quite substantially in their life cycles and physiology and may share some immune mechanisms, but may also have evolved different mechanisms to sense and defend against viruses. This review will compile the current knowledge of how viral PAMPs are sensed by mosquitoes and what we know and suspect from other organisms, and we will highlight potential gaps in our current understanding of mosquitoes’ immune sensing of viruses. 

## 2. Dicer2 and DExD/H RNA Helicases

Antiviral immunity, and the recognition of viruses in particular, through the RNAi pathway is the most well-understood of all the pathways involved in the arthropod antiviral immune response. The RNAi pathway is one of the major innate immune pathways that control virus infection in insects. Insect RNAi comprises three distinct pathways mediated by either short interfering RNAs (siRNAs), microRNAs (miRNA), or PIWI-interacting RNAs (piRNAs) [16]. Viral recognition through and the activation of the piRNA and miRNA pathways remains unclear, but virus-derived piRNAs are generated in multiple mosquito species to a number of viruses [17,18,19], and in silico analysis indicates that cellular mosquito miRNAs may recognize and bind directly to viral genomes [20]. Further, the exogenous siRNA pathway is clearly activated during virus infection in mosquitoes (and other arthropods) in response to arbovirus infection. This subject has been reviewed extensively in the past [14] so we will not go into detail here. The only step we would like to draw attention to is the first step in the RNAi response, which is viral double-stranded RNA (dsRNA) recognition [21]. 

Viral dsRNA is a major PAMP in the immune response of any organism. Mammalian Rig-I-like receptors (RLRs) are well-established sensors of viral dsRNA that promote an interferon response [22,23,24,25]. In insects, the endonuclease Dicer-2 (Dcr-2) is related to mammalian RLRs and also recognizes viral dsRNA, but processes it into virus-derived siRNAs [26,27]. The correct dsRNA processing and RNAi functioning of *Drosophila* and mosquito Dcr-2 is dependent on a DExD/H RNA helicase domain [28,29,30,31]. Proteins with this domain fall into a large superfamily of RNA helicases [32,33]. Out of all the DExD/H RNA helicases, *Drosophila* Dcr-2’s helicase domain is most closely related to the RLR helicase domain, and is also required for the upregulation of the *vago* gene following infection with Drosophila C virus (DCV) and Sindbis virus (SINV) [34]. Vago is a small cysteine-rich secreted polypeptide that acts like a cytokine, comparable to an interferon response [34]. Importantly, Vago was crucial for controlling DCV replication in *Drosophila* fat bodies [34]. These findings suggested an RNAi-independent role of Dcr-2 in insect antiviral immunity that resembles RLR signaling in mammals [35]. It was also observed that the recombinant Flock House virus protein B2, which binds dsRNA, and thus, inhibits Dcr-2 function, reduced *vago* expression in flies following infection with DCV, further implicating that Dcr-2 signaling through *vago* expression plays an antiviral role in response to dsRNA [34,36]. Similar to human RLRs [37], the helicase domain of Dcr-2 in flies and mosquitoes was found to be responsible for sensing defective viral genomes, which feed into viral DNA synthesis and boost the siRNA response [35,38]. Whether there is intracellular immune signaling associated with this sensing is unknown. 

In mosquitoes, Dcr-2 has been shown to activate the NF-κB transcription factor Rel2 in *Culex quinquefasciatus* Hsu cells in response to West Nile virus (WNV) infection [39,40]. Following the activation of Rel2, the expression of *vago* was upregulated, leading to the activation of the JAK/STAT pathway (see Section 5) and the expression of antiviral genes [39,40]. This finding established Vago as an antiviral cytokine-like molecule in mosquitoes. *Vago* expression is also induced in *Aedes albopictus* cells after dengue virus (DENV) infection, although this was not shown to be dependent on Dcr-2 [40]. In addition, *vago* upregulation was not observed in two studies that treated *Aedes aegypti* Aag2 cells with a dsRNA mimic, poly(I:C) [41,42]. However, *vago* upregulation may require nucleic acid features that are unique to viral dsRNA or may depend on the exact site of virus replication [34]. Altogether, there is evidence that dsRNA acts as a viral PAMP for mosquito Dcr-2 sensing that leads to the activation of RNAi-independent immune signaling pathways, but it remains to be seen whether this is conserved across mosquito species and which viruses are impacted by this immune response. 

Mosquitoes also encode DExD/H RNA helicases that differ phylogenetically from the RLR subfamily, including DEAD- and DEAH-box helicases [43,44]. Many of these are responsible for sensing viral infection in other organisms [45,46,47,48] and some have been implicated in antiviral responses in insects (Table 1). For instance, in *Drosophila,* DDX56 binds CHIKV RNA and limits infection [49]. Taschuk et al. also found five other DEAD-box helicases in *Drosophila* that were antiviral against SINV [49]. Recent efforts have also implicated select *Ae. aegypti* helicases in the response to virus infection. The *Ae. aegypti* DEAD-box helicase ME31B, an ortholog of mammalian DDX6, was recently shown to bind the subgenomic flavivirus RNA (sfRNA) of WNV and ZIKV and to exhibit antiviral activity against these viruses in *Ae. aegypti* Aag2 cells [50]. However, ME31B binding to sfRNA was not shown to initiate an immune response. It was proposed that sfRNA binding actually enhances viral replication by sequestering ME31B, thereby counteracting its antiviral role [50]. In *Drosophila*, ME31B is antiviral against La Crosse virus (LACV) and Rift Valley fever virus (RVFV) via its canonical role of depleting the available pool of host capped mRNAs available for cap snatching [51]. In humans, DDX6 was found to be a co-sensor with RIG-I for influenza virus and enterovirus RNA that mediated the antiviral interferon response [52,53]. DDX6 has also been implicated in a proviral state for several arboviruses in human cells [48]. While ME31B has clearly been indicated in mosquito and fly antiviral responses, it remains unclear whether it acts as a classic PRR and, if so, which downstream antiviral responses are activated. 

Another DEAD-box helicase in *Drosophila*, Rm62, binds bunyaviral RNA and restricts LACV and RVFV replication but not SINV or vesicular stomatitis virus (VSV) in cells and flies [54]. The binding of RVFV RNA by the human ortholog DDX17 appears not to trigger interferon signaling, but restricts replication directly [54]. The *Ae. aegypti* ortholog of Rm62 was found to bind CHIKV nsP3 in Aag2 cells [55], but the consequences of this interaction need to be investigated further. The antiviral effect of human DDX17 has so far only been suggested to be a result of viral RNA degradation, either alone or as a co-factor with the zinc-finger antiviral protein (ZAP) [54,56]. The putative role of mosquito Rm62 in antiviral defenses and virus sensing remains unclear, but existing evidence warrants further investigation.

The *Ae. aegypti* ortholog of mammalian DHX15, a DEAH-box helicase, is antiviral against CHIKV, although this appears to be dependent on its function in regulating glycolysis as opposed to sensing virus infection directly [57]. However, there is considerable evidence that human DHX15 acts as a PRR for dsRNA in response to virus infection [46]. In addition to DHX15, Machado et al. identified two other broadly acting antiviral DExD/H-box helicases in mosquitoes [57]. One of these is an ortholog of human DHX9, which is known to sense viral nucleic acids and can lead to a pro- or antiviral outcome depending on the virus and host context [46]. 

Another *Ae. aegypti* DExH domain-containing protein, Spindle-E, is antiviral against two alphaviruses in Aag2 cells [58]. Spindle-E also contains a Tudor domain and was thus hypothesized to play a role in the RNAi response, like other Tudor domain-containing proteins [59]. Yet, its antiviral activity was independent of the RNAi response, suggesting other antiviral mechanisms that may include dsRNA sensing and immune signaling [58]. 

Considering their ability to sense viral nucleic acids and proteins in other organisms, and current evidence from mosquitoes, DExD/H-box RNA helicases may represent an important and understudied class of PRRs in mosquitoes that warrant additional research. 

## 3. The IMD Pathway

The immunodeficiency (IMD) pathway is an inducible signaling pathway that leads to the activation of the NF-κB transcription factor Rel2 (Relish in *Drosophila*) and subsequent antimicrobial peptide (AMP) production [60,61,62]. It has been well studied for its involvement in non-viral pathogen defenses, which typically are mediated through the peptidoglycan-recognition protein (PGRP) recognition of bacteria and fungi in *Drosophila* [63,64,65]. In *Anopheles gambiae*, the IMD pathway regulates *Plasmodium* and bacterial infection [66]. For *Plasmodium* defense, PGRP-LC, PGRP-LA, and PGRP-S2/PGRP-S3 all play a role in *Anopheles coluzzi*, while PGRP-LB promotes permissiveness to infection [67]. Rel2 and other IMD pathway components protect *Ae. aegypti* from Gram-negative and Gram-positive bacteria [62,68]. The antiviral properties of the *Drosophila* IMD pathway have also been demonstrated [69]. Conflicting reports about whether the IMD pathway plays an antiviral role in mosquitoes currently exist (reviewed in [8]). It is also worth noting that Rel2 may not be strictly linked to just the IMD pathway. As previously described, Dcr-2 can also activate Rel2, seemingly independently of IMD [34,39,40].

It has been found that microbiota present in the midgut of *Ae. aegypti* increase Rel2 levels [70]. Thus, studying the putative antiviral effects of Rel2 may be confounded by changes in the bacterial microbiome, since blood feeding can trigger nutrient-induced growth in microbiota and the associated Rel2 upregulation [70]. IMD’s involvement in the antiviral response may, therefore, depend, in part, on bacterial PRRs. In line with this, it has been shown that heat-killed bacteria can restore the IMD-dependent antiviral defense of flies that have been treated with antibiotics [71]. Additionally, *pgrp-lc* mutant flies have increased ZIKV and DCV loads after virus challenge, further confirming the role of bacterial sensing in antiviral responses [71,72]. 

Yet, some direct virus-induced activation and upregulation of IMD pathway components has been observed. In *Ae. aegypti* mosquitoes, blood-fed SINV increased Rel2 expression compared to a non-infectious control, suggesting the presence of a virus-specific receptor that can upregulate an important component of the IMD pathway [70]. Further, Sansone et al. demonstrated that DCV protection in flies is dependent on both bacterial and viral stimulation of the IMD pathway [71]. Another study showed that the knockdown of PGRP-LC had no effect on the replication of a SINV replicon in flies, while other IMD pathway components did have an effect [73]. However, Barletta et al. depleted Caspar, the negative regulator of Rel2, as a method of activating the IMD pathway, and this led to decreased microbiota and increased SINV levels [70]. These findings further highlight the importance of bacterial contributions and microbiome balance in antiviral immunity [70]. Finally, one of the previously mentioned studies using poly(I:C) to mimic dsRNA stimulation showed the upregulation of genes downstream of IMD and Rel2 signaling in Aag2 cells [41]. While dsRNA sensing may not signal directly through IMD, it can be sensed by other proteins (e.g., Dcr-2) that activate Rel2, bypassing the canonical IMD pathway. In line with this, a recent paper showed that the RVFV infection of *Ae. aegypti* Aag2 cells can prime immune responses, resulting in increased expression of certain bacterial PRRs and an enhanced response to bacterial challenge [74]. These data indicate that RVFV infection is indeed sensed by the mosquito immune system, resulting in increased IMD pathway sensitivity. 

## 4. Toll Receptors

In mammals, the function of Toll-like receptors (TLRs) as PRRs for a wide array of ligand types from many different pathogens is relatively well understood [75]. Mammalian TLRs can sense a variety of microbial PAMPs, including those of viruses. TLR3 is perhaps the most studied in terms of virus infection as its ectodomain binds dsRNA, leading to a pro-inflammatory immune response [76,77,78]. It has now been documented that TLRs 2, 3, 6, 7, 8, 9, and 10 all recognize viruses to some extent, either through foreign nucleic acid sensing or the detection of viral surface proteins [75,79,80]. 

The canonical immune responsive Toll pathway in *Drosophila* functions through the detection of a PAMP by a PRR that triggers cleavage of the extracellular ligand Spaetzle, which then binds to and activates the transmembrane Toll-1 receptor [81,82]. Similar to the IMD pathway, this cascade, in turn, leads to the activation of the NF-κB transcription factors Dorsal and Dif in *Drosophila*, or Rel1A/B in mosquitoes, and AMP expression [83,84,85]. Bacterial and fungal PRRs that are known to activate Spaetzle and Spaetzle-like proteins include Gram-negative binding protein 1, PGRP SA, and Gram-positive specific serine protease [86,87]. A Toll receptor in *Ae. aegypti*, Toll5a, recently had its structure solved and was shown to bind to Spaetzle1C [88]. Spaetzle-like proteins themselves have been shown to contribute to antiviral immunity against several viruses in *Drosophila*, often by upregulating AMP expression [89,90,91]. However, it is still unclear which PRR is responsible for the cleavage of Spaetzle-like proteins in response to viral infection in flies or mosquitoes. Nonetheless, several studies have shown other Toll pathway components to be antiviral in *An. gambiae* and *Ae. aegypti*, although specific Toll receptors or ligands have not been discovered [92,93,94]. 

*Drosophila* Toll-1 and Toll-7 have been shown to bind VSV directly, which actually resulted in the induction of the autophagy pathway [89,95]. A Toll receptor from the insect vector *Laodelphax striatellus* bound Rice-Stripe virus nucleocapsid protein and contributed to antiviral defenses, although this did not seem to be dependent on other canonical Toll-pathway components either [96]. Additionally, Toll-7 was required for the autophagy-dependent restriction of RVFV in *Drosophila* [97]. However, others have found that Toll-7 is not required for *Drosophila* resistance to VSV and autophagy plays a minor role [98]. Toll-1 and Toll-7 binding to VSV and Spaetzle-like proteins did, however, result in AMP promoter activation [89]. Further complicating the determination of the source of activation and which pathway is induced, a study found that, in *Ae. aegypti*, *Wolbachia*-mediated reactive oxygen species activated the Toll pathway and led to the upregulation of AMPs and the inhibition of DENV replication [99]. 

The Toll gene family has expanded in most mosquito species compared *Drosophila* [61,100]. For example, two mosquito species have experienced Toll-9 gene duplications [100]. In addition, mosquitoes have gained two new clades, Toll10 and 11, while losing two others, Toll2 and 3. Inferences of function based on homology alone may not be sufficient for these Toll genes, especially those that have undergone taxa-specific expansion. There is a need to determine Toll gene function experimentally [101]. Of course, homology inquiries have been used as launching points for successful studies. For example, based on in-depth phylogenetic analysis and homology modeling of human TLR4, *Bombyx mori* Toll-9 was studied and shown to recognize lipopolysaccharides [102]. Another recent study found that a dsRNA binding site of TLR3 is conserved in *Ae. aegypti* Toll6 (AaToll6) [42]. Importantly, AaToll6 was upregulated in Aag2 cells after treatment with poly(I:C) [42]. Together, these findings were used to suggest that AaToll6 is a PRR-recognizing dsRNA. In another study, however, poly(I:C) did not activate the Rel1A promoter region in Aag2 cells, and instead, induced Rel2-dependent expression [41]. The discrepancy in these two studies may be due, in part, to experimental differences, such as the method of poly(I:C) delivery [103]. For instance, in mammals, TLR3 primarily senses dsRNA in endosomes while RLRs sense it in the cytoplasm [104]. The delivery and location of dsRNA within the cell may thus impact the type of immune response that is elicited. Overall, evidence for AaToll6 or any other Toll receptor directly interacting with a viral PAMPs in response to virus infection in mosquitoes is still lacking. It is likely that, in some cases, upstream viral PRRs are responsible for activating the Toll pathway in mosquitoes, perhaps through Spaetzle. 

## 5. The JAK/STAT Pathway

In *Drosophila,* the JAK/STAT signaling pathway begins with the activation of Unpaired (Upd) ligands (Upd, Upd2, Upd3) that signal through the receptor Domeless (Dome). How (and if) Upd ligands are activated in response to viral infection is still unknown, but it may be mediated by insulin signaling (see Section 6) [105]. Ultimately, the pathway concludes with the activation of the transcription factor STAT and downstream antiviral defenses, depending on the virus/insect combination (reviewed in [84]). In mosquitoes, Upd ligands have not yet been identified [8]. However, the cytokine-like molecule Vago was shown to activate the JAK/STAT pathway in *Cx. Quinquefasciatus* Hsu cells, with downstream antiviral effects on WNV replication [39]. As discussed in Section 2, *vago* expression was upregulated after WNV infection via Dcr-2/Rel2 signaling, implicating Dcr-2 as a PRR for JAK/STAT pathway activation. However, the receptor Dome was not necessary for anti-WNV action, prompting the question of whether another JAK/STAT receptor exists [39]. Recently, a study on kuruma shrimp (*Marsupenaeus japonicus*) identified integrin as a receptor for Vago that was essential for antiviral activity against white spot syndrome virus (WSSV) [106]. This study also identified ficolin as the antiviral effector molecule upregulated following Vago-mediated JAK/STAT signaling that bound the virus and promoted its clearing [106]. However, in this instance, Dcr-2’s involvement in vago upregulation was not investigated. It is unknown what other mosquito PRRs are responsible for activating the JAK/STAT pathway in response to viral infection, although recent work has identified some possible mechanisms (see Section 7). Insulin may also play a role (see Section 6). 

## 6. MAPKs

MAPKs are involved in signaling cascades that have recently emerged as components of mosquito antiviral immunity (reviewed in [8]). In mammals, a diverse set of stimuli can activate MAPKs, which, in turn, regulate the activity of target proteins, including transcription factors [107]. Two conventional MAPKs include extracellular signal-regulated kinase (ERK) and c-Jun N-terminal kinase (JNK), which are conserved in mosquitoes [108]. Many reports have now demonstrated that the ERK pathway is induced by virus infection in mosquitoes, serves an antiviral role, and, importantly, can be enhanced by the mammalian insulin present in a bloodmeal [109,110,111]. Ahlers et al. showed that Upd and JAK/STAT pathway activation was dependent on insulin-mediated ERK pathway activation [110]. The microbiota-dependent stimulation of the IMD pathway, discussed above, was also shown to lead to the activation of the ERK pathway in flies [71]. These reports point towards the stimulation of ERK being largely dependent on non-viral stimuli. 

Interestingly, Liu et al. showed that silencing JNK in *Ae. Albopictus* C6/36 cells had no impact on DENV2 RNA levels, while another study saw increased DENV2, ZIKV, and CHIKV RNA levels after silencing Kayak, a JNK pathway core component, in *Ae. Aegypti* salivary glands [109,112]. This discrepancy may be due to a species-specific response and/or represent a difference between cell-culture and in vivo studies. Nonetheless, how the JNK pathway is activated remains largely unknown. In *Drosophila*, the JNK pathway can be activated through the PGRP-LC recognition of peptidoglycan [113]. However, similar activation is not observed in C6/36 cells [114]. Chowdhury et al. proposed that the antiviral JNK pathway activity they observed was due to infection-induced oxidative stress [112]. Whether this is the extent of JNK activation in response to viral infection or whether any direct virus sensing can activate JNK remains to be seen. 

## 7. TEPs and LRRs

The complement system has long been shown to be a crucial component of vertebrate innate immunity [115]. At its core, the C3 protein, one of the most abundant circulating plasma proteins, is involved in all three complement activation pathways, which all lead to opsonization and contribute to the clearing of pathogens [116,117]. C3, along with additional complement factors, is part of the thioester-containing protein (TEP) family. These proteins share a conserved and highly reactive thioester CGEQ amino acid motif [118,119]. In vertebrates, there are two subfamilies of TEPs—complement factors and the closely related α2-macroglobulins, which also serve a role in innate immunity [120]. An additional TEP subfamily specific to insects was described through studies on *Drosophila.* These complement-like insect TEPs (iTEPs) were identified based on high sequence similarity to the C3/α2-macroglobulin family [121]. One of the roles of these iTEPs may be as PRRs upstream of the Toll pathway, as indicated by reduced Toll signaling after microbial infection in flies lacking all four of the secreted iTEPs [122]. 

A mosquito iTEP, *An. gambiae* TEP1 (AgTEP1), has been characterized in detail and shown to be of high structural and functional similarity to vertebrate C3 [123]. Functional analysis showed the ability of AgTEP1 to recognize, directly bind, and promote the clearing of Gram-negative bacteria and *Plasmodium* parasites [124,125]. The antiviral activity of iTEPs was showcased in mosquitoes through the silencing of *Ae. aegypti* TEP1 (AaTEP1), which led to a significant increase in WNV and DENV2 load [126]. A more recent study confirmed AaTEP1 to have anti-DENV activity in *Ae. aegypti* mosquitoes and suggested that this activity was the result of enhanced AMP production [127]. Interestingly, in these TEP1-depleted transgenic mosquitoes, *rel2* expression, but not *rel1* expression, was suppressed compared to wild-type mosquitoes following a bloodmeal [127]. These results indicate a possible link between AaTEP1 and the IMD pathway, although Rel2 expression can be driven by enhanced microbiota growth after a bloodmeal [70] (see Section 3 above). These results also further highlight the complex interactions between the mosquito immune response, the microbiome, and virus replication. Another TEP, AaTEP20, impacted ZIKV replication and AMP levels when silenced in mosquitoes [112]. In none of these studies, however, were AaTEPs investigated directly for their role in sensing viruses. 

Insects all possess a varying number of iTEP genes, possibly suggesting an evolutionary expansion ba sed on pathogen-specific recognition needs [128]. Similarly, a repertoire of host proteins with a leucine-rich repeat (LRR) motif contribute to the recognition of variable PAMPs in animals and plants and even comprise the foundation for adaptive immunity in agnathans [129,130]. This LRR domain is conserved across many families of immune proteins and species [131]. Copies of these motifs form the structural basis of mammalian TLRs, facilitate TLR pathogen binding in humans, and are also present in mosquito and tick Toll receptors [132,133,134,135]. The AgTEP1 binding of *Plasmodium* was found to be dependent on two LRR immune proteins (LRIM 1 and 2) [129,136,137]. In silico analyses later revealed many LRR-containing genes in mosquito disease vectors that share similar features with LRIM 1 and 2, and therefore, were proposed to represent a diversified set of PRRs in mosquitoes [138]. Differential gene expression analysis showed two LRR-containing genes to be downregulated compared to other immune genes after infection with ZIKV and CHIKV in *Ae. aegypti* [139]. In addition, an LRR-containing gene was enriched upon challenge with DENV in a genetic DENV-refractory strain of *Ae. aegypti*, a change that perhaps confers DENV restriction [139,140].

An RNAi screen revealed the *Drosophila* macroglobulin/complement-related subfamily (DmMCR) as another functionally significant class of TEPs in insects. DmMCR was found to bind *Candida albicans* and promote phagocytosis in *Drosophila* [141]. Although often grouped under iTEPs, phylogenetic analysis reveals unique structural properties of MCRs [119]. Most notably, DmMCR (AKA DmTEP6) and its mosquito orthologs AgMCR (AKA AgTEP13) and AaMCR (AKA AaTEP13) all have a mutated thioester motif [61,128,142]. In contrast to iTEPs, MCRs are highly conserved among arthropods [61]. Together, these differences suggest a mechanism of pathogen recognition separate from iTEPs. AaMCR was shown to bind the E protein of DENV via an adaptor molecule, an *Ae. aegypti* homolog of scavenger receptor-C (AaSR-C) (see Section 8), suggesting another mechanism of arbovirus recognition by mosquito TEPs and related proteins [142]. The recognition of DENV by AaSR-C and AaMCR was also shown to regulate the expression of AMPs exerting anti-DENV activity, directly linking virus recognition to an antiviral signal transduction immune response [142]. Importantly, the AMPs regulated by the AaMCR/AaSR-C system share an overlap with Toll and JAK/STAT-regulated genes [142,143].

## 8. Scavenger Receptors

Scavenger receptors are a large family of heterogeneous molecules that are well-established to recognize danger-associated molecular patterns and PAMPs in other organisms, as well as carry out functions such as binding lipoproteins [144] (reviewed in [145,146]). They have been divided into 10 classes based on structural homology [147,148]. Class-C SRs consist of one member that was first described in *Drosophila* and is absent in mammals [148,149]. Since its discovery, it has become recognized as a PRR, primarily for Gram-negative and Gram-positive bacteria [150]. However, as stated above, an *Ae. aegypti* homologue was discovered and found to bind directly to DENV [142]. Studies on other arthropods have also identified SR-C orthologs, including in the kuruma shrimp, where it binds WSSV and promotes its clearance [151]. Another type of SR, class-A1, was recently described and shown to bind CHIKV’s NSP1 via Co-IP in mice, which facilitated virus control by inducing autophagy [152]. In addition, the E2 protein of multiple alphaviruses can be bound by the mouse scavenger receptor SR-A6 (MARCO) to induce the clearing of a circulating virus independent of the complement system or natural antibodies [153].

Multiple scavenger receptors are conserved across mosquito species genomes, and it would be worth investigating whether any have antiviral activity [100,154]. So far, only a few SRs besides DmSRC have been studied in *Drosophila*, and only in the context of antibacterial immunity (reviewed in [12]). It is important to note, however, that there are also cases of viruses using SRs to facilitate cellular entry [145]. For example, a high-density lipoprotein receptor, SR-B1, is responsible for internalizing ZIKV and DENV by binding the NS1 of both viruses in human Huh7 and *Ae. albopictus* C6/36 cells [155]. 

Besides promoting the autophagy of infected cells and being involved in the phagocytosis of microbes, including viruses [145,151], the overexpression of SR-C in *B. mori* was recently found to upregulate Toll pathway components and AMPs [156]. Indeed, it has been known for some time that SRs interact with TLRs in mammals as co-receptors required for NF-κB signaling in response to bacterial infections [157,158,159]. Additionally, SR-A1 is expressed on the surface of human epithelial cells and can bind dsRNA to mediate its uptake and downstream signaling [160]. Later, it was also shown that SR-A1 is needed for the RLR and TLR3 sensing of dsRNA [161]. In this case, SR-A1 is thought to act as a gatekeeper that delivers dsRNA to the established receptors inside the cell [161]. Ultimately, it was shown that SR-A1 is necessary for the TLR3-mediated antiviral response to hepatitis C virus [162]. Mosquitoes encode five class-A SRs, representing an interesting avenue for future research [100]. 

## 9. Lectins

Calcium-type (C-type) lectins (CTLs) describe a large group of proteins that bind carbohydrates in a calcium-dependent manner, although many CTL domains (CTLDs) have been found in proteins that do not bind sugar and/or depend on calcium [163,164]. CTLD-containing proteins are known to play a key role in the vertebrate immune system by acting as PRRs for a multitude of pathogens [165]. Mannose binding lectin (MBL), a secreted CTLD-containing protein, is responsible for recognizing and binding to N-linked glycans on WNV structural proteins in mice [166]. In addition, an ELISA assay showed MBL’s ability to bind DENV virions in humans as a crucial step in viral clearance [167].

In insects, two novel CTLs derived from the housefly, *Musca domestica*, exhibited significant antiviral activity in lepidopteran S9 cells [168]. However, while some mosquito CTLs (mosCTL) have antibacterial [169] and antifungal [170] properties, evidence of any mosCTL antiviral activity is lacking. In contrast, as has been seen above with other innate immune receptors, most mosCTLs that have been studied for their involvement in viral immunity act to facilitate infection [171], suggesting that viruses have evolved to hijack these proteins traditionally involved in immunity. In 2010, Cheng et al. identified mosquito galactose-specific binding CTL-1 (mosGCTL-1) in *Aedes* mosquitoes as crucial for WNV infection after gene silencing, based on an RNAi screen of human proteins, and found that blood feeding with mosGCTL-1 antisera reduced vector competence [172,173]. WNV attachment to cells via mosGCTL-1 also required the membrane-bound mosPTP-1, an ortholog of human CD45 [172]. A similar effect with a *Cx. quinquefasciatus* mosGCTL-1 homolog was seen [172]. However, a more recent study failed to interfere with WNV infection using anti-mosGCTL-1 antibodies in *Culex pipiens* [174]. Additional mosCTLs have been shown to bind the viral envelopes of DENV and JEV, promoting their entry into mosquito cells [175,176]. Further advances in *Ae. aegypti* genome annotation have uncovered over 50 CTLs, a significant portion of which have unknown function in the mosquito immune response to viral infection [171]. It is thus entirely possible that other mosCTLs exhibit antiviral activity through virus binding and viral PAMP sensing.

## 10. Phenoloxidase Activity

Melanization is a key innate immune mechanism in mosquitoes and other invertebrates that is managed by the highly regulated phenoloxidase (PO) enzyme and results in hemocytes’ production of melanin, promoting pathogen clearance [177,178]. However, in some cases, immune defense is achieved without the production of melanin [179]. In some insect species, namely the moth *Manduca sexta*, the serine protease cascade that leads to PO activation has been studied at length [178,180]. Molecules involved in the early stages of PO activation have thus been elucidated. The hemolymph protease 14, for instance, binds bacteria directly and senses fungal infection through beta-1,3-glucan recognition proteins (GRP) [181,182,183,184,185]. Two CTLs in *Drosophila* seem to act as bacterial PRRs that promote melanization [186]. In a genetically selected strain of *An. gambiae*, melanization is responsible for refractoriness to *Plasmodium*, mediated by TEP1 activity [187]. PRRs leading to PO activation have been identified in mosquitoes, including a mosquito GRP capable of binding multiple pathogens in *Armigeres subalbatus* [188]. 

The activation of the PO pathway by insects in response to virus infection is also well documented. An early study showed that *Heliothis virescens* larval hemolymph demonstrated antiviral activity towards Sindbis virus in a PO- and melanization-dependent manner [189]. Virucidal activity against a baculovirus was a consequence of H_2_O_2_ produced from the PO activation cascade in *Heliothis virescens* [190]. Interestingly, a poly DNA virus carried by the parasitoid wasp *Microplitis demolitor* encodes a protein, Egf1.0, that acts as a serine proteinase inhibitor (serpin) to block the PO cascade, although this may be responsible for blocking anti-parasite activity as opposed to antiviral PO activity [191]. There are now numerous documented pathogen factors that act to block proPO activation in invertebrates (reviewed in [179]). Together, these findings point to a crucial role of PO activity in insect immunity, with its antiviral roles less clearly defined. 

In conditioned media from *Ae. albopictus* U4.4 cells, a PO cascade resulted in decreased viability of the arbovirus Semliki Forest Virus (SFV), and this decrease was correlated with increased melanization activity [192]. In the same study, recombinant SFV expressing Egf1.0 inhibited U4.4 PO activity and enhanced virus spread in cell culture and in *Ae. aegypti* mosquitoes [192]. Whether TEPs, lectins, or other receptors are responsible for sensing viral infection in mosquitoes and triggering the proPO activation cascade is unknown. In *Drosophila*, crosstalk between the IMD, Toll, and PO pathways has been observed in response to bacterial challenge, further complicating the interpretation of studies into all of these antimicrobial pathways [193,194].

## 11. Conclusions and Future Research

Over the last two decades, significant advances have been made in the field of virus recognition in mosquitoes, encompassing studies on a variety of innate signaling pathways and humoral responses. Despite these efforts, a comprehensive picture with clear connections from the initial virus detection to the signaling and clarification of pathway crosstalk is still missing. Some information is derived from *Drosophila*, some from *Ae. aegypti*, and some from *Cx. quinquefasciatus* or other mosquitoes. Similarly, most studies understandably focus on one specific virus as opposed to employing a more broadly comparative approach. Few studies have looked at the first steps of virus recognition—which mosquito PRRs detect which viral PAMPs and trigger mosquito immune responses. There is a need for comparative immunology studies across both *Aedes* spp. and *Culex* spp. mosquitoes and multiple arbovirus families. We have compiled the various potential mechanisms by which viruses may be sensed in mosquitoes, highlighted potential crosstalk between pathways, and discussed the complex nature of microbiome-mediated immunity versus virus-induced immunity. All of the immune responses outlined above require further studies to improve our current understanding of how antiviral responses are elicited and how these mechanisms differ between various viruses and mosquito species. With the advancement of bioengineering techniques, including gene editing in mosquitoes [195,196,197] and progress in gene drive strategies [198,199], it has become more evident that we need to fully understand the fundamental virus–mosquito interactions to design the best possible targets for gene editing. Disruption of the immune system has already been shown to increase disease phenotypes in mosquitoes following infection [200], while transgenic activation of the JAK/STAT pathway reduced DENV virus replication in vivo [201]. However, the better our understanding of mosquito immune responses, the better we can predict the success and limitations of gene targets for arbovirus transmission control. 

## Figures and Tables

**Figure 1 biomolecules-13-01159-f001:**
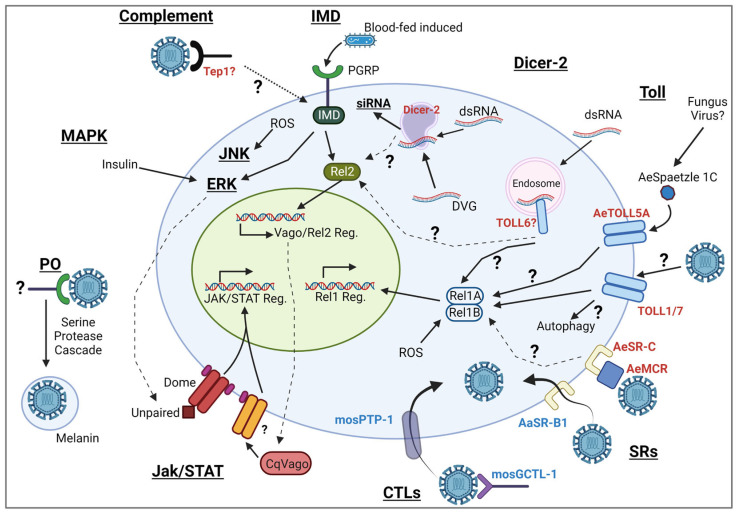
Mosquito antiviral immune pathways. An overview of the different antiviral defense strategies available to mosquitoes and the potential receptors that act as pattern recognition receptors to initiate them. We focus on receptors for which there is experimental evidence for their putative role in sensing virus infection, including *Drosophila melanogaster* and mosquitoes. Receptors that are suspected or shown to be responsible for sensing viruses directly and initiating an antiviral response are colored in red. Protein receptors that may facilitate viral entry are colored in blue. Question marks denote strong candidate receptors or signaling pathways that so far lack any direct evidence for the receptor function in mosquitoes. Distinct pathways are in bold/underlined. Dotted arrows represent canonical pathway crosstalk. Created using BioRender.com (accessed on 16 June 2023).

**Table 1 biomolecules-13-01159-t001:** *Ae. aegypti* DExD/DExH-box helicases with putative PRR or antiviral activity.

Name/ Accession	Motif II Sequence	Antiviral Activity	Binding Partner	Human Ortholog
**ME31B** **(AAEL008500)**	DEAD	Yes	sfRNA	DDX63 #
Rm62 * (AAEL008738)	DEAD	Unknown	CHIKV nsP3	DDX17 #
**DHX15 (AAEL004419)**	DEAH	Yes	Unknown	DHX15 #
AAEL008728	DETD	Yes	Unknown	DDX24
**AAEL004859**	DEIH	Yes	Unknown	DHX9 #
Spindle E (AAEL013235)	DEIH	Yes	Unknown	TDRD9

* Antiviral PRR in *Drosophila.* # Antiviral PRR in mammals.

## Data Availability

No new data were created or analyzed in this study. Data sharing is not applicable to this article.

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
