# Peer review of "Recognition of Arboviruses by the Mosquito Immune System"

_biomolecules, 2023, doi:10.3390/biom13071159_

Round 1
Reviewer 1 Report
The authors produced a comprehensive and up-to-date review covering the various mechanisms, pathways, molecules and receptors involved in the immune defense of insects against pathogens, extrapolating, whenever possible, to mosquitoes and arboviruses. The work is well written and can help other scientists looking for the "state of the art" in this field of knowledge. Some suggestions are described below.
Line 31: In my opinion mosquitoes are not "causative agents" but the vectors of these agents. Please consider replacing.
Line 35: Why did the authors mention an order (Bunyavirales) and two families instead of mentioning the three (or more) viral families (Bunyaviridae, Flaviviridae and Togaviridae)?
Lines 48 to 55: Please, insert bibliography to support this information.
Lines 83 to 93: Please consider citing the interesting works of Dr. Pei-Shi yen, on miRNA.
Line 135: Please, correct the scientific name by using lower case to “aegypti”.
Line 249 to 252: The use of mosquitoes infected with Wolbachia has been shown to be one of the most promising tools for controlling Aedes aegypti and arboviruses. Perhaps the authors could review more deeply how the presence of these bacteria modulate the immune system of mosquitoes.
Line 256: Toll-10?
Line 340: Is it possible to determine which DENV serotype was tested, as mentioned above (line 311 for example)?
Line 365: “Aedes aegypti”
Line: Please, check if “lepidopteran” should be italicized.
Author Response
We thank both reviewers for their time and thoughtful comments. All line numbers below refer to the marked up revised version of the manuscript. Aside from addressing the reviewer’s comments where appropriate, we have also added two sentences to include a very recent reference that we had previously missed (lines 233-238) which shows that RVFV can prime an IMD mediated immune response.
Reviewer 1:
The authors produced a comprehensive and up-to-date review covering the various mechanisms, pathways, molecules and receptors involved in the immune defense of insects against pathogens, extrapolating, whenever possible, to mosquitoes and arboviruses. The work is well written and can help other scientists looking for the "state of the art" in this field of knowledge. Some suggestions are described below.
Line 31: In my opinion mosquitoes are not "causative agents" but the vectors of these agents. Please consider replacing.
We apologize for missing this poor wording – we agree, they transmit the causative agents and are not causative agents themselves. This has been reworded.
Line 35: Why did the authors mention an order (Bunyavirales) and two families instead of mentioning the three (or more) viral families (Bunyaviridae, Flaviviridae and Togaviridae)?
We understand the confusion, but with the current nomenclature introduced about 5 years ago, there is no family ‘Bunyaviridae’ anymore. The Bunyavirales include many families, 3 of which are families definitely including arboviruses (Peribunyaviridae, Phenuiviridae, and Nairoviridae). We felt it was easier to combine these 3 families in an inclusive ‘Bunyavirales’ approach. Other sources have switched to referring to the order instead of the different families as well.
Lines 48 to 55: Please, insert bibliography to support this information.
We have added two citations (reviews) to support this information.
Lines 83 to 93: Please consider citing the interesting works of Dr. Pei-Shi yen, on miRNA.
While this work is generally based on computer prediction, not directly showing virus recognition by cellular miRNAs, we have added a sentence referring to it (lines 97-99).
Line 135: Please, correct the scientific name by using lower case to “aegypti”.
Thank you for catching this – we have fixed it in multiple instances.
Line 249 to 252: The use of mosquitoes infected with Wolbachia has been shown to be one of the most promising tools for controlling Aedes aegypti and arboviruses. Perhaps the authors could review more deeply how the presence of these bacteria modulate the immune system of mosquitoes.
Within the context of our review, we do not think that it is appropriate to expand on bacterial interactions with the immune system. We mention bacteria and other factors a few times to highlight how these can be confounding our studies on virus-sensing or specific examples of interactions, but we do not wish go into detail about any particular bacteria.
Line 256: Toll-10?
We have decided to use Toll10 and Toll6 etc. (without the dash) for mosquito Toll nomenclature in this review. We only used the dash were referring to Drosophila Tolls.
Line 340: Is it possible to determine which DENV serotype was tested, as mentioned above (line 311 for example)?
Yes, thank you. We have added the serotype (DENV2) for this instance as well (see new line 363).
Line 365: “Aedes aegypti”
Thank you for catching this – we have fixed it in multiple instances.
Line: Please, check if “lepidopteran” should be italicized.
It should not, we apologize for this oversight and have remove the italics.
Reviewer 2 Report
Dear Authors,
I have reviewed your manuscript "Recognition of arboviruses by the mosquito immune system", submitted for publication in Biomolecules.
I find that your manuscript is a high-quality contribution in that it comprehensively and systematically brings together the current state of knowledge on this important topic. The English language of the manuscript is very good. The abundant informations are at times a bit condensed, requiring a very dedicated focus on the part of the reader, for which I generally recommend "dilution" of parts of text with elaborations, short explanations and similar, wherever you identify an appropriate point (but see some examples below). The manuscript could benefit from some clarifications, additional summarizations, as well as from putting the current state of knowledge into an engineering perspective. Please follow my recommendations for the revision of your manuscript as given below:
· Introduction:
o the addition of a Table, summarizing the most important arboviral diseases, citing, for instance, the name of the disease, the viral agent, the mosquito host, and perhaps some epidemiological features (such as, for example, the concerned part of the world, most common symptoms, year of first, or most recent, or historically most notable outbreak) might be useful to the readers. This is in no way mandatory, but at least a list of the most important arboviral diseases and concerned viruses and mosquitos would very much refresh the Introduction.
o line 31, first sentence: This sentence is incorrect - mosquitos are not the causative agents, but vectors for these diseases.
o line 41-43: One of the factors that is recently increasingly cited as contributing to the outbreaks of novel diseases, is the disruption of natural habitats, which brings viruses and other pathogens that had previously dwelled "in the middle of a jungle" in close proximity of areas densely populated with humans. This argument has been abundantly brought up in relation to SARS-CoV-2, but I believe that it can also apply to arboviral diseases.
o line 60: Please fully spell out "PRRs" within the Figure caption, although it is already fully spelled out in the manuscript text. A Figure caption is supposed to be able to stand alone independently of the manuscript text.
o It would be beneficial to state, at some point within the Introduction, that the knowledge on many of these immune mechanisms in mosquitos is still quite scarce, for which knowledge obtained on other dipterans such as Drosophila melanogaster, and even other insects, needs to be discussed as an indicator of the most likely ways in which the yet insufficiently researched mosquito immune mechanisms work.
· Dicer2:
o The Section 2 is very complex and quite difficult to follow at moments. The Authors should consider a possibility of creating a second figure, dedicated specifically to the content of Section 2, but more detailed than Figure 1. Of course, other similar Figures can be created in other sections where the complexity of narration is too demanding for a reader.
o line 104: "comparable to interferon an interferon response" - please double-check this sentence
· Sections 3-10:
o line 182: please fully spell out "AMP" at first mention.
o line 189: have also been demonstrated
o line 200: I believe that, talking about flies with a mutation in the PGRP-LC gene, we should call them "pgrp-lc mutants". Please double-check. Also please thoroughly double-check the entire manuscript for the writing style of mutant and mutation names.
o line 215: The names of the genes IMD and Rel2 should be put into italic. In this sentence, you are clearly talking about the genes, and not their protein products. Please thoroughly revise the entire manuscript to put all the gene names always in italic (not necessary when you are talking about the corresponding proteins).
o line 246: I believe it should read RVFV and not RFVF.
o lines 253-256: This part of the text sounds as if mosquitos had evolved from Drosophila. Please revise to accurately reflect the fact that both Drosophila and mosquitos have evolved from a common ancestor (which might be more related to Drosophila, than to the mosquitos, but it is still not Drosophila itself).
o line 310: Please specify, for the readers who are not familiar with mosquitos, which mosquito species the cells C6/36 belong to. In the next sentence, you mention possible species-specific differences between what was found in C6/36 and Aedes aegypti, so the species identity of C6/36 should be mentioned.
o line 343: Rel2 expression (Rel2 in italic, because expression refers to a gene) was suppressed after a bloodmeal in comparison to before the bloodmeal, or in comparison to wild-type mosquitos after a bloodmeal? "Suppressed after a bloodmeal" intuitively sounds like "in comparison to before the bloodmeal", but from the context I think you meant in comparison with wild-type mosquitos after the bloodmeal. Please be clear about this.
o line 349-350: Were they unable to prove that AaTEP is involved in virus sensing, or did they seem to show that AaTEP is not involved? This is another example where your text could benefit from clearer statements.
o line 427: Why is "lepidopteran" written in italic?
o line 429-onwards: If this makes sense, I would like to see a commentary here, that, from the evolutionary perspective, the CTLs proved prone to manipulation by viruses, whereby the viruses were able to evolve mechanisms by which they could manipulate and transform CTLs (which had previously evolved as part of immune mechanisms) into a vehicle for more efficient penetration into the host cells.
o line 465: I believe that you should shorten "antiviral activity of PO activity" to "antiviral PO activity".
· Article Conclusion - Reading your article's Abstract, I was looking forward to learn what kind of engineering approaches might be envisaged to disrupt the balance between the mosquito immune mechanisms and the viral replication, the balance which, as you said in the Abstract, is crucial to the role of mosquitos as arbovirus vectors. Although in your article you described in great detail the mechanisms of virus recognition by the mosquito immune system, I was somewhat disappointed to find that this engineering perspective was lacking within the article itself. I understand that the current state of research is probably "not there yet", and that more knowledge needs to be collected before such engineering endeavors can be undertaken. However, I would like you to clearly say something about that in the Conclusion to your article. I wouldn't change anything within your Abstract - I find very exciting that you offered this kind of perspective within the Abstract (and besides, it was important to underline the importance of the topic of your review). I would just want you to reiterate, in the Conclusion section, this engineering perspective as a possible optimistic goal of the future research, and maybe offer a brief commentary about "where we're at" right now with respect to that perspective, i.e., what are the gaps that we yet need to fill so that we can better understand the relationship between mosquitos and arboviruses, and how to manipulate that relationship to end the role of mosquitos as virus vectors.
In the end, I want to congratulate you on the hard work behind your manuscript. I am looking forward to read its published version in Biomolecules.
Kind regards,
Reviewer
Author Response
We thank both reviewers for their time and thoughtful comments. All line numbers below refer to the marked up revised version of the manuscript. Aside from addressing the reviewer’s comments where appropriate, we have also added two sentences to include a very recent reference that we had previously missed (lines 233-238) which shows that RVFV can prime an IMD mediated immune response.
I find that your manuscript is a high-quality contribution in that it comprehensively and systematically brings together the current state of knowledge on this important topic. The English language of the manuscript is very good. The abundant informations are at times a bit condensed, requiring a very dedicated focus on the part of the reader, for which I generally recommend "dilution" of parts of text with elaborations, short explanations and similar, wherever you identify an appropriate point (but see some examples below). The manuscript could benefit from some clarifications, additional summarizations, as well as from putting the current state of knowledge into an engineering perspective. Please follow my recommendations for the revision of your manuscript as given below:
We thank the reviewer for the kind words. We have addressed comments below as/where appropriate or provided explanation of our rationale. We have also added a short section about the use of bioengineering in the conclusion as requested below, but we do consider it outside the scope of this review to go into more detail. We do not wish to draw away too much specifically from ‘sensing’ of virus infection.
Introduction:
the addition of a Table, summarizing the most important arboviral diseases, citing, for instance, the name of the disease, the viral agent, the mosquito host, and perhaps some epidemiological features (such as, for example, the concerned part of the world, most common symptoms, year of first, or most recent, or historically most notable outbreak) might be useful to the readers. This is in no way mandatory, but at least a list of the most important arboviral diseases and concerned viruses and mosquitos would very much refresh the Introduction.
We thank the reviewer for raising this concern, and while this would not be difficult to add, this information is so abundant in other resources/reviews (e.g. reference 2) that it seems redundant here. We tried to keep a specific focus on mosquito immune recognition in our review and we anticipate that the topic and the special issue that this review will be part of (‘New Insight into Vector Borne Diseases’) will draw in an audience that is generally familiar with arboviruses.
line 31, first sentence: This sentence is incorrect - mosquitos are not the causative agents, but vectors for these diseases.
We apologize for missing this poor choice of wording – we agree, they transmit the causative agents and are not causative agents themselves. This has been reworded.
line 41-43: One of the factors that is recently increasingly cited as contributing to the outbreaks of novel diseases, is the disruption of natural habitats, which brings viruses and other pathogens that had previously dwelled "in the middle of a jungle" in close proximity of areas densely populated with humans. This argument has been abundantly brought up in relation to SARS-CoV-2, but I believe that it can also apply to arboviral diseases.
We agree and have added this additional factor to the sentence in line 42.
line 60: Please fully spell out "PRRs" within the Figure caption, although it is already fully spelled out in the manuscript text. A Figure caption is supposed to be able to stand alone independently of the manuscript text.
Thank you for bringing this to our attention. We have spelled it out.
It would be beneficial to state, at some point within the Introduction, that the knowledge on many of these immune mechanisms in mosquitos is still quite scarce, for which knowledge obtained on other dipterans such as Drosophila melanogaster, and even other insects, needs to be discussed as an indicator of the most likely ways in which the yet insufficiently researched mosquito immune mechanisms work.
We thank the reviewer for this suggestion – we have added such a statement in lines 80-86 as we are setting up the premise of this review.
Dicer2:
The Section 2 is very complex and quite difficult to follow at moments. The Authors should consider a possibility of creating a second figure, dedicated specifically to the content of Section 2, but more detailed than Figure 1. Of course, other similar Figures can be created in other sections where the complexity of narration is too demanding for a reader.
While we thank the reviewer for the suggestion, there really is hardly any additional detail that could be integrated into a figure here. We already include Dicer-2 sensing and the signaling via Rel2 in Figure 1 and there is no further concrete evidence for other mechanisms aside from TRAF4 involvement. We would be recreating a section of Figure 1. We have listed the various DexD/H helicases in the table, because we do not know if and how they fit into other signaling mechanisms, so a figure is not feasible, but we do think some of them directly recognize foreign dsRNA and are relevant to mention here.
We appreciate that this section is the hardest to follow, but this in part simply due to the many open questions related to dsRNA sensing in mosquitoes.
line 104: "comparable to interferon an interferon response" - please double-check this sentence
Thank you for bringing this to our attention. We have fixed this duplicated wording.
Sections 3-10:
line 182: please fully spell out "AMP" at first mention.
Thank you, we fixed this and corrected the text accordingly.
line 189: have also been demonstrated
Thank you, we fixed this in the revised version.
line 200: I believe that, talking about flies with a mutation in the PGRP-LC gene, we should call them "pgrp-lc mutants". Please double-check. Also please thoroughly double-check the entire manuscript for the writing style of mutant and mutation names.
Thank you for catching this. We apologize – it can be hard to consistently distinguish between the gene and protein. We double checked this in a few instances and have addressed these based on your suggestions where appropriate.
line 215: The names of the genes IMD and Rel2 should be put into italic. In this sentence, you are clearly talking about the genes, and not their protein products. Please thoroughly revise the entire manuscript to put all the gene names always in italic (not necessary when you are talking about the corresponding proteins).
We are in fact talking about the proteins here. We mention the genes downstream of IMD and Rel2, meaning that IMD and Rel2 are referred to as the proteins acting as immune factors and transcription factors in the signaling process. To clarify this, we added the word ‘signaling’ in the sentence for clarity (now line 231).
line 246: I believe it should read RVFV and not RFVF.
Yes, thank you for catching this typo. We fixed it in the revised version.
lines 253-256: This part of the text sounds as if mosquitos had evolved from Drosophila. Please revise to accurately reflect the fact that both Drosophila and mosquitos have evolved from a common ancestor (which might be more related to Drosophila, than to the mosquitos, but it is still not Drosophila itself).
Thank you for catching this. We removed the wording about ‘expansion from the original Drosophila Tolls’ and instead tried to rephrase it as a comparison of parallel evolution without indicating that mosquitoes evolved from Drosophila itself. (Lines 275-278)
line 310: Please specify, for the readers who are not familiar with mosquitos, which mosquito species the cells C6/36 belong to. In the next sentence, you mention possible species-specific differences between what was found in C6/36 and Aedes aegypti, so the species identity of C6/36 should be mentioned.
We have added ‘Ae. albopictus’ to the statement to clarify the cell origin.
line 343: Rel2 expression (Rel2 in italic, because expression refers to a gene) was suppressed after a bloodmeal in comparison to before the bloodmeal, or in comparison to wild-type mosquitos after a bloodmeal? "Suppressed after a bloodmeal" intuitively sounds like "in comparison to before the bloodmeal", but from the context I think you meant in comparison with wild-type mosquitos after the bloodmeal. Please be clear about this.
We have fixed this to rel2 (in italics) since you are of course right and expression refers to the gene. We also rephrased the sentence as shown below, for clarity:
“Interestingly, in these TEP1 depleted transgenic mosquitoes, rel2 expression, but not rel1 expression, was suppressed compared to wild-type mosquitoes following a bloodmeal’. We can see how (if determined) this could still be misinterpreted somehow, but is clearer than the previous version. (lines 366-368)
line 349-350: Were they unable to prove that AaTEP is involved in virus sensing, or did they seem to show that AaTEP is not involved? This is another example where your text could benefit from clearer statements.
We appreciate the confusion and have substituted the statement with the following: “In none of these studies, however, were AaTEPs investigated directly for their role in sensing viruses.” (lines 374-375)
line 427: Why is "lepidopteran" written in italic?
We apologize for this oversight and have removed the italics.
line 429-onwards: If this makes sense, I would like to see a commentary here, that, from the evolutionary perspective, the CTLs proved prone to manipulation by viruses, whereby the viruses were able to evolve mechanisms by which they could manipulate and transform CTLs (which had previously evolved as part of immune mechanisms) into a vehicle for more efficient penetration into the host cells.
We think this is a bit outside the scope and not a rare phenomenon – many viruses in other systems use proteins/receptors involved in immunity or on immune cells for virus entry. We added a short sentence to address this in line 456-457.
line 465: I believe that you should shorten "antiviral activity of PO activity" to "antiviral PO activity".
Yes, thank you. We realize now that it sounds a bit awkward and have changed it accordingly.
Article Conclusion - Reading your article's Abstract, I was looking forward to learn what kind of engineering approaches might be envisaged to disrupt the balance between the mosquito immune mechanisms and the viral replication, the balance which, as you said in the Abstract, is crucial to the role of mosquitos as arbovirus vectors. Although in your article you described in great detail the mechanisms of virus recognition by the mosquito immune system, I was somewhat disappointed to find that this engineering perspective was lacking within the article itself. I understand that the current state of research is probably "not there yet", and that more knowledge needs to be collected before such engineering endeavors can be undertaken. However, I would like you to clearly say something about that in the Conclusion to your article. I wouldn't change anything within your Abstract - I find very exciting that you offered this kind of perspective within the Abstract (and besides, it was important to underline the importance of the topic of your review). I would just want you to reiterate, in the Conclusion section, this engineering perspective as a possible optimistic goal of the future research, and maybe offer a brief commentary about "where we're at" right now with respect to that perspective, i.e., what are the gaps that we yet need to fill so that we can better understand the relationship between mosquitos and arboviruses, and how to manipulate that relationship to end the role of mosquitos as virus vectors.
I see now what you are referring to when you mentioned the ‘engineering perspective’ above. While we think it is beyond the scope to go into detail on this subject, we have added a few sentences to our conclusion addressing this as a future perspective (lines 515-523).
In the end, I want to congratulate you on the hard work behind your manuscript. I am looking forward to read its published version in Biomolecules.
Thank you for your overall positive review and critical feedback.